# CPLLM: Clinical Prediction with Large Language Models

## Abstract

We present Clinical Prediction with Large Language Models (CPLLM), a method that involves fine-tuning a pre-trained Large Language Model (LLM) for clinical disease prediction. We utilized quantization and fine-tuned the LLM using prompts, with the task of predicting whether patients will be diagnosed with a target disease during their next visit or in the subsequent diagnosis, leveraging their historical diagnosis records. We compared our results to various baselines, including Logistic Regression, RETAIN, and Med-BERT, which is the current state-of-the-art model for disease prediction using temporal structured EHR data. Our experiments have shown that CPLLM surpasses all the tested models in terms of PR-AUC and ROC-AUC metrics, displaying noteworthy enhancements compared to the baseline models.

## 1 Introduction

Large Language Models (LLMs) are a type of artificial intelligence (AI) that have been shown to be effective at a variety of Natural Language Processing tasks (Zhao et al., 2023). LLMs are trained on large amounts of textual data, which allows them to learn the statistical relationships between words and phrases. LLMs are used for different types of tasks, including natural language understanding, natural language generation, knowledge-intensive tasks, reasoning, and more (Yang et al., 2023). This makes them well-suited for tasks that require understanding the meaning of text, such as text classification (Gasparetto et al., 2022; Sun et al., 2023) and even clinical predictions in the medical domain (Thirunavukarasu et al., 2023; Steinberg et al., 2021).

Clinical predictions are used for estimating a patient's susceptibility to disease, gauging the likelihood of treatment response, or prognosticating the course of a patient's medical condition. (Laupacis et al., 1997; Wasson et al., 1985). These predictions have been executed via classical models such as Logistic Regression (Hosmer Jr et al., 2013) and Random Forest (Breiman, 2001). However, these traditional methods do not model the order of the medical concept events (diagnoses, procedures, medications, etc.). Instead, they rely solely on the absence or presence of these events (features).

Modern event order prediction models, which are more advanced than the mentioned traditional prediction models, are based on RNNs or transformers, where the latter were shown to be superior (Vaswani et al., 2017). Specifically, BERT-Style Models like BERT (Devlin et al., 2018), RoBERTa (Liu et al., 2019), and Deberta (He et al., 2020). Another transformer-based architecture is GPT-style language model. GPT models are trained to generate the next word in a sequence. GPT models are used in a wide range of downstream tasks such as summarization, translation, question answering and more (Floridi & Chiriatti, 2020). To name a few GPT models: LLaMA (Touvron et al., 2023a;b), Falcon (Almazrouei et al., 2023), Bloom (Scao et al., 2022), and GPT4 (OpenAI, 2023). The flexibility and versatility of decoder-only models seem to be advantageous (Yang et al., 2023).

The significance of the mentioned language models for handling sequential data is emphasized, particularly within the context of clinical prediction models relying on Electronic Health Record (EHR) data. Structured EHR data encompasses a patient's clinical history, notable for its irregular temporal sequence of events and observations (Steinberg et al., 2021). Previous works deal with modeling EHR diagnosis data as a sequence, such as BEHRT (Li et al., 2020; 2022a; Shoham & Rappoport, 2023; Meng et al., 2021), Med-BERT (Rasmy et al., 2021) and Medic-BERT (Hansen et al., 2023) (for length of stay prediction), using BERT models. However, they represent each

diagnosis code as an index and do not address the text description of the ICD code. In addition, they did pre-training based on clinical data and they have limited sequence length according to the BERT architecture.

There is limited research on the use of LLMs to train clinical prediction models. One of the main focus of applications of LLM in the clinic is on the chat capability of these models (Singhal et al., 2023; Thirunavukarasu et al., 2023) or using an LLM for medical texts (Lu et al., 2022; Sivarajkumar & Wang, 2022; Li et al., 2022b; Jiang et al., 2023; Agrawal et al., 2022; Yang et al., 2022). In addition, Chen et al. (2023) proposed a method called ClinTaT for cancer prediction. Their focus was on cancer prediction using few-shot learning, and their data modeling was not designed for structured EHR data that consists of a sequence of diagnoses. However, we want to harness the power of LLMs in understanding sequences of tokens derived from structured EHR data, specifically to train prediction models. We represent the structured data as a text by representing each medical concept corresponds to a word, admissions are treated as visits, and patient history is considered a document. The objectives of this study are to develop a novel method for using LLMs to train clinical predictors and to evaluate the performance of this method on real-world datasets.

Our proposed method uses an LLM to predict future diagnoses of patients by fine-tuning LLMs. The medical concepts are represented by text descriptions. Fine-tuning is performed using a prompt that feeds the model with training samples. We used two different LLMs, Llama2, which is a general LLM (Touvron et al., 2023b) and BioMedLM which was trained on biological and clinical text (Venigalla et al., 2022). We used three prediction tasks and two datasets and compared the performance to three baseline models.

The proposed method outperforms the state-of-the-art methods. Our generic method can be used for a variety of tasks and is not specific to any particular LLM. Moreover, our method is also suitable for different clinical domains such as demographics, diagnoses, laboratory test results, measurements, procedures, and more.

**Contributions**: **(1)** We propose CPLLM, a novel method for clinical prediction with LLM that outperforms state-of-the-art models for disease prediction for structured EHR data. In addition, CPLLM doesn't require pre-training on clinical data and achieved better performance. Moreover, Our method has a longer sequence length limit compared to the baseline methods. **(2)** We show that adding additional tokens to the pre-trained tokenizer of the LLM before fine-tuning improves the performance of the clinical prediction model. **(3)** Our code is flexible for any LLM, available to use, and easily adaptable to various diagnosis prediction tasks.

## 2 METHODS

### 2.1 DISEASE PREDICTION - PROBLEM DEFINITION

Formally, for a given patient $p$, let $n$ denote the total number of diagnoses in their medical history. Thus, the patient's sequence of diagnoses is represented as $\{D_{p,1}, D_{p,2}, D_{p,3}, \ldots, D_{p,n}\}$, where each $D_{p,i}$ $(1 \leq i \leq n)$ corresponds to a medical diagnosis in the patient's history. We considered two types of diagnosis prediction: next diagnosis and next visit diagnosis.

**Next diagnosis prediction**: Predict whether patient $p$ will be diagnosed with a specific disease $D_x$ as the $D_{p,i+1}$ diagnosis given previous diagnoses. Our model relys on the patient's medical records up to the $i$-th diagnosis, denoted as $\{D_{p,1}, D_{p,2}, \ldots, D_{p,i}\}$. Where $D_{p,i}$ $(1 \leq i < n)$ indicates the most recent diagnosis observed for patient $p$. The predictive model utilizes this patient-specific historical medical information to determine whether patient $p$'s next diagnosis is a specific disease or not.

**Next visit diagnosis prediction**: Predicting the next diagnosis requires knowledge of the precise timing of each diagnosis. However, these data may occasionally be unavailable, such as when diagnoses are documented at the end of an admission. Consequently, in the context of the MIMIC-IV dataset, we undertake the task of forecasting whether a patient will receive a specific diagnosis in his subsequent admission.

## 2.2 DATA

In this study, we used data from the eICU-CRD database (Pollard et al., 2018) and data from the MIMIC-IV database (Johnson et al., 2020). Our datasets include ICD-9-CM (eICU-CRD) and ICD-10-CM (MIMIC-IV) diagnoses and their descriptions. In the eICU-CRD database, each diagnosis is associated with a timestamp. Consequently, we arranged the diagnoses in chronological order based on their respective diagnosis times. Our disease prediction task aims to anticipate whether the forthcoming diagnosis will correspond to a specific disease. Unlike the eICU-CRD dataset, the MIMIC-IV data lacks information on the time of each diagnosis. However, it provides the start time for admission and the discharge times for each patient. As a result, our prediction task for this dataset revolves around determining whether a patient will be diagnosed with a specific disease during his subsequent visit.

Med-BERT adopts a pre-training strategy and trains BERT using Masked Language modeling and Length of stay (LOS) prediction tasks (Rasmy et al., 2021). Therefore, we extracted the necessary data from the databases, including the ICD diagnosis codes for each patient. Additionally, we also include information on the LOS of each admission and the number of visits of each patient. On the other hand, In our approach, we did not conduct an additional pre-training step, as we focused on fine-tuning a Large Language Model (LLM). In our proposed method, it's not required to note at which visit each diagnosis was given. Furthermore, the duration of hospital stay is not required. Notably, our method attains superior results even in the absence of these particulars. This aspect holds significance, since in certain situations, this data may not be accessible. For example, when a patient has not been admitted to the hospital but is under the care of a family doctor.

**Data Preprocessing**: For the MIMIC-IV dataset we excluded patients with only one visit, as there is no medical history in such a case. Similarly, for the eICU-CRD dataset, patients with just one diagnosis were removed. We also excluded patients who have the disease we are trying to predict at the first visit (or the first diagnosis for eICU-CRD data). We converted our ICD-10 codes to their corresponding Clinical Classification Software (CCS) categories for MIMIC-IV, while for eICU-CRD, we retained the ICD-9 codes as they were. This decision was motivated by the higher number of ICD-10 codes compared to ICD-9 codes (Manchikanti et al., 2013). Based on the sequence of diagnoses for each patient, we determined whether the patient exhibited a specific diagnosis based on ICD diagnosis codes related to the specific disease according to the relevant CCS category (Elixhauser et al., 2014). Table 1 provides an overview of the number of patients, the count of final patients after preprocessing, average diagnoses, and average visits for each disease prediction task.

### 2.2.1 CLINICAL OUTCOMES

We conducted an evaluation of our model using data related to three distinct diseases: Chronic kidney disease and Acute and unspecified renal failure predictions, both derived from the MIMIC-IV dataset, and Adult respiratory failure prediction sourced from the eICU-CRD dataset. The corresponding CCS codes for these diseases are 157 for Acute and unspecified renal failure, 158 for Chronic kidney disease, and 131 for Adult respiratory failure. For each prediction task, patients with specific disease ICD codes were assigned a positive label, and their diagnosis history encompassed all diagnostic codes recorded until the specific code indicated the outcome of interest. For instance, for patient $p$, when predicting whether this patient would develop specific disease in the future, given a sequence of ICD-9 codes $\{D_{p,1}, \ldots, D_{p,n}\}$, and the patient was diagnosed with specific disease at index $i$, the patient history was represented as $\{D_{p,1}, \ldots, D_{p,i-1}\}$, and the label was positive. Conversely, for a patient without a diagnosis of the outcome of interest, we randomly selected an index $j$ ($1 \leq j \leq n$) using a uniform distribution, and the patient history was defined as $\{D_{p,1}, \ldots, D_{p,j}\}$, resulting in a negative label. Through this approach, we created labeled data for our predictive model, allowing us to perform the task of forecasting whether patients would be diagnosed with specific diseases based on their medical history.

Table 1: Task statistics of the prediction tasks. Visit and diagnosis counts are calculated from the patient's medical history after preprocessing. IQR - Interquartile range.

| Dataset | Task | # of patients | Final # of patients | Median # of visits (IQR) | Median # of diagnoses (IQR) |
|---------|------|---------------|---------------------|--------------------------|------------------------------|
| MIMIC-IV | Chronic kidney disease | 84,453 | 26,161 | 1 (1-2) | 11 (7-19) |
| MIMIC-IV | Acute and unspecified renal failure | 84,453 | 26,736 | 1 (1-2) | 11 (7-19) |
| eICU-CRD | Adult respiratory failure | 132,677 | 56,419 | 1 (1-1) | 1 (1-2) |

## 2.3 BASELINE METHODS

we conducted a rigorous performance assessment of the CPLLM against a set of three baseline methods. First, Med-BERT with a classification layer (Rasmy et al., 2021). Second, with Logistic Regression (Hosmer Jr et al., 2013). Furthermore, we compared our method to RETAIN, a disease prediction model featuring double GRUs and attention modules (Choi et al., 2016). We compared CPLLM with these baseline methods to gain valuable insights into its performance in clinical prediction downstream tasks. The comparison was conducted using two metrics: the area under the precision-recall curve (PR-AUC) and the area under the receiver operating characteristic curve (ROC-AUC). Disease prediction tasks are typically imbalanced, and as Davis & Goadrich (2006) pointed out, ROC-AUC is less suitable for imbalanced datasets for binary classification. Therefore, our main evaluation metric is PR-AUC, but we also report ROC-AUC for consistency with the baseline methods.

## 2.4 OUR PROPOSED METHOD

We propose a method called Clinical Prediction with Large Language Models (CPLLM). This method involves fine-tuning a Large Language Model (LLM) using prompts tailored to medical concept sequences. Through fine-tuning using prompts (inputs for LLM guidance), we direct the LLM to grasp intricate relationships among medical concepts.

We utilized two LLMs: Llama2 (13B parameters) (Touvron et al., 2023b) and BioMedLM (also called PubMedGPT, 2.7B parameters) (Venigalla et al., 2022). To enhance the time and memory efficiency of fine-tuning these LLMs, we used QLoRA (Dettmers et al., 2023) and PEFT (Houlsby et al., 2019). QLoRA is a PEFT approach that decreases the number of parameters requiring fine-tuning and also performs quantization (Dettmers et al., 2023). This combined approach effectively optimized the models' efficiency, enabling single-GPU fine-tuning for both BioMedLM and Llama2 models.

We performed separate fine-tuning of each LLM, leveraging specific prompts tailored to our patients' diagnoses and their corresponding labels. In Figure 1, we present the prompts utilized during the fine-tuning process for both the Llama2 and BioMedLM. We also indicated in the prompt the target disease, and the prompts were designed to incorporate the patients' individual diagnoses histories, with the goal of improving the models' performance. We added new tokens to the vocabulary of the LLM tokenizer, in order to prevent the LLM from not knowing the descriptions of our diagnosis codes. We performed an ablation study that compared the performance with and without changing the vocabulary of the pre-trained tokenizer, in the Experiments section.

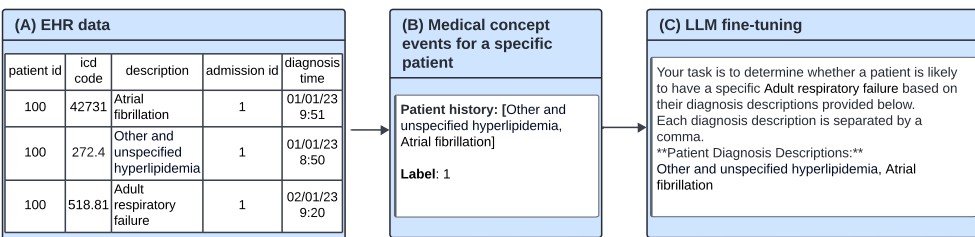

Figure 1: Illustration of the fine-tuning process. (A) illustrates an example of EHR structural data. (B) represents the patient history extracted from the EHR data, specifically for patient id 100, where the diagnosis history is represented through descriptions. (C) shows the fine-tuning procedure applied to LLMs. Fine-tuning prompts consist of a general description, the patient's diagnosis history, and a label. The label is set to 1 when the patient is diagnosed with Adult Respiratory Failure in the subsequent diagnosis or during the next admission, depending on the task.

For the clinical prediction downstream task, we performed fine-tuning on both Llama2 and BioMedLM, as depicted in Figure 1. We used prompts to ask the LLMs to generate a single binary token in response (1 or 0) by adding a classification layer corresponding to the number of labels.

By training the models with all patients' data for the specified number of epochs, we obtained the fine-tuned LLM tailored to our specific clinical prediction task.

# 3 EXPERIMENTS

## 3.1 EXPERIMENTAL SETUP

We compare our method to three baseline models. The first is a simple Logistic Regression that does not model the data as a sequence but as simple independent, unordered variables (Manogaran & Lopez, 2018). The second is RETAIN which is a two-level neural attention model (Choi et al., 2016). The third baseline is Med-BERT, which is the state-of-the-art for structured EHR data for disease prediction. RETAIN was the baseline of Med-BERT. We split our data using an 80-10-20 ratio, where 80% for the train set, 10% for the validation set, and the last 20% for the train set. For Med-BERT, we trained the pre-training model with the MLM and LOS tasks, with the TensorFlow package (Abadi et al., 2015). The training of the Med-BERT's MLM phase was performed according to the fixed number of steps in the original implementation. The training took about 1.5 days on an RTX1080 GPU. Subsequently, we performed fine-tuning on the pre-trained model for the specific clinical prediction downstream tasks. The RETAIN and Med-BERT baselines trained for 500 epochs with early stopping based on the PR-AUC value derived from the validation set, using a maximum number of epochs without improvement of 5 epochs (Prechelt, 2002). During the training of the baselines, we experimented with various batch sizes $\{32, 100\}$ and different learning rates $\{1e^{-5}, 2e^{-5}\}$. For each prediction task, we selected the hyper-parameters that achieved the best results on the validation set. For Logistic Regression training, we utilized the scikit-learn package (Pedregosa et al., 2011) and trained the model on a CPU. To determine the optimal hyper-parameters for Logistic Regression, we conducted a grid search encompassing *penalty* (L1 and L2 regularization), *C*, *solver*, and the maximum number of iterations. We explored values of $\{0.1, 1, 10\}$ for *C*, {'liblinear', 'saga'} for *solver*, and $\{100, 200, 500\}$ for the number of iterations. We took the best hyper-parameters based on the validation PR-AUC for each prediction task.

For CPLLM experiments, we utilized two large language models called Llama2 (13B) and BioMedLM (2.7B) and fine-tuned them using the HuggingFace library (Wolf et al., 2019). To optimize the fine-tuning process, we used Parameter-Efficient Fine-Tuning (PEFT) (Houlsby et al., 2019; Mangrulkar et al., 2022) and QLoRA (Dettmers et al., 2023). Specifically, we used a learning rate of $2e^{-5}$, Lora alpha of 32, Lora dropout of 0.1, and bias of none. Given the resource constraints, we meticulously determined and employed the maximum batch size that our GPU memory could accommodate. We fine-tuned each model over six epochs, selecting the best checkpoint based on validation PR-AUC. Fine-tuning Llama2 for six epochs required about a day of training on an RTX 6000 GPU, while BioMedLM took about two hours on the same hardware. Our fine-tuning process used FEPT, and we didn't perform additional pre-training in the clinical domain, yet our CPLLM method outperformed the baseline models.

## 3.2 RESULTS

We consider various models for the clinical prediction task: Logistic Regression, Med-BERT with a classification layer, RETAIN, and our proposed method called CPLLM. To examine the statistical significance of the results, we ran each model three times. Table 2 shows the mean and 95% confidence interval of PR-AUC and ROC-AUC of these models.

Our findings demonstrate that our method, CPLLM, outperforms all tested models, including RE-TAIN, Med-BERT, and Logistic Regression, across both PR-AUC and ROC-AUC metrics. Specifically, in the context of the Adult respiratory failure task, CPLLM-Llama2 achieved a noteworthy PR-AUC value of 35.962%, signifying an absolute improvement of 0.912% over the best-performing baseline model, Logistic Regression, which obtained a PR-AUC score of 35.05%. This improvement corresponds to a relative enhancement of 2.6% in PR-AUC. Additionally, our method exhibits a relative increase of 5.1% in PR-AUC when compared to RETAIN and a 3.31% increase when compared to Med-BERT. Regarding ROC-AUC performance, CPLLM outperforms the baseline models. Furthermore, CPLLM-Llama2 demonstrates superior performance in this specific task compared to CPLLM-BioMedLM. Logistic Regression outperforms RETAIN in both PR-AUC (35.05%) and

ROC-AUC (74.664%), but it also outperforms Med-BERT in PR-AUC, albeit not in ROC-AUC (74.664% compared to 75.407% for Med-BERT).

For Chronic kidney disease using the MIMIC-IV dataset, RETAIN had the worst performance in both metrics. Med-BERT outperformed Logistic Regression and RETAIN. CPLLM-Llama2 had the highest PR-AUC score of 33.992%, followed by CPLLM-BioMedLM with 33.984% and Med-BERT with 33.37%. However, in ROC-AUC, CPLLM-BioMedLM outperformed all models with a score of 83.404%, followed by CPLLM-Llama2 with 83.034% and Med-BERT with 83.12%.

For Acute and unspecified renal failure, CPLLM-Llama2 achieved the highest measurements, boasting a PR-AUC score of 45.442% and an ROC-AUC score of 78.504%. This signifies a notable improvement of 4.22% in PR-AUC compared to the leading baseline model, RETAIN, in this task. Additionally, it demonstrates a 1.31% improvement in ROC-AUC compared to the best-performing baseline, which is Logistic Regression with an ROC-AUC score of 77.486%. Furthermore, it's worth highlighting that in this specific task, RETAIN outperforms Med-BERT in terms of PR-AUC but not ROC-AUC. Additionally, CPLLM-Llama2 demonstrates superior performance compared to CPLLM-BioMedLM.

| Task | Model | PR-AUC | ROC-AUC |
|---|---|---|---|
| Adult respiratory failure | Logistic Regression | 35.050 | 74.664 |
| | RETAIN | $34.22 \pm 0.299$ | $74.454 \pm 0.173$ |
| | Med-BERT | $34.81 \pm 0.208$ | $75.407 \pm 0.073$ |
| | CPLLM-Llama2 | $\mathbf{35.962 \pm 0.380}$ | $\mathbf{76.407 \pm 0.262}$ |
| | CPLLM-BioMedLM | $35.494 \pm 0.352$ | $75.975 \pm 0.214$ |
| Chronic kidney disease | Logistic Regression | 32.230 | 83.016 |
| | RETAIN | $31.407 \pm 1.379$ | $81.692 \pm 0.899$ |
| | Med-BERT | $33.37 \pm 0.891$ | $83.12 \pm 0.173$ |
| | CPLLM-Llama2 | $\mathbf{33.992 \pm 1.262}$ | $83.034 \pm 0.511$ |
| | CPLLM-BioMedLM | $33.984 \pm 1.077$ | $\mathbf{83.404 \pm 0.429}$ |
| Acute and unspecified renal failure | Logistic Regression | 42.075 | 77.486 |
| | RETAIN | $43.603 \pm 0.409$ | $77.364 \pm 0.394$ |
| | Med-BERT | $42.237 \pm 0.408$ | $77.427 \pm 0.185$ |
| | CPLLM-Llama2 | $\mathbf{45.442 \pm 0.839}$ | $\mathbf{78.504 \pm 0.684}$ |
| | CPLLM-BioMedLM | $45.161 \pm 1.622$ | $78.484 \pm 0.403$ |

Table 2: Performances of various models assessed across multiple tasks and datasets, highlighting the PR-AUC and ROC-AUC metrics. Higher is better.

## 3.3 ABLATION STUDY

We conducted an ablation study to investigate the impact of the added tokens to the pre-trained tokenizer of the LLMs before fine-tuning. Table 3 provides a comprehensive overview of the PR-AUC and ROC-AUC, comparing scenarios with and without the addition of extra tokens. For the task of predicting Acute and unspecified renal failure, adding the tokens yields enhancements in both PR-AUC and ROC-AUC for CPLLM-Llama2 (0.499% absolute increase in PR-AUC and a 0.554% absolute increase in ROC-AUC). Similarly, CPLLM-BioMedLM shows substantial improvements with a 1.631% absolute increase in PR-AUC, representing a relative enhancement of 3.746%, and a 0.414% absolute increase in ROC-AUC. In contrast, for the prediction of Chronic kidney disease, the inclusion of extra tokens does not significantly impact PR-AUC and ROC-AUC in the case of CPLLM-Llama2. However, CPLLM-BioMedLM demonstrates improvements, specifically an absolute enhancement of 0.686% in ROC-AUC and an increase in PR-AUC from 32.638% to 33.984%. It is worth noting that the PR-AUC of BioMedLM exhibits less stability, as evidenced by a larger confidence interval when no additional tokens are employed (4.358%). Nevertheless, we conducted two additional runs to get a better estimate of the PR-AUC. Subsequently, we observed that the PR-AUC for these five experiments amounted to 33.078%, and the confidence intervals were reduced to 1.773%. For Adult respiratory failure prediction, the presence of additional tokens results in improved PR-AUC and ROC-AUC for CPLLM-Llama2, whereas it enhances PR-AUC but does not influence ROC-AUC for CPLLM-BioMedLM. In summary, the findings of this ablation study

suggest that, in the majority of cases (9 out of 12 measurements across three prediction tasks), the incorporation of the added tokens leads to enhanced performance in clinical prediction tasks.

| Task | Model | Added Tokens | PR-AUC | ROC-AUC |
|------|-------|--------------|--------|---------|
| Acute and unspecified renal failure | CPLLM-Llama2 | yes | **45.442 ± 0.839** | **78.504 ± 0.684** |
| | | no | 44.943 ± 1.268 | 77.95 ± 0.814 |
| | CPLLM-BioMedLM | yes | **45.161 ± 1.622** | **78.484 ± 0.403** |
| | | no | 43.53 ± 1.101 | 78.07 ± 0.625 |
| Chronic kidney disease | CPLLM-Llama2 | yes | 33.992 ± 1.262 | 83.034 ± 0.511 |
| | | no | **34.563 ± 1.578** | **83.178 ± 1.02** |
| | CPLLM-BioMedLM | yes | **33.984 ± 1.077** | **83.404 ± 0.429** |
| | | no | 32.638 ± 4.358 | 82.718 ± 1.191 |
| Adult respiratory failure | CPLLM-Llama2 | yes | **35.962 ± 0.38** | **76.407 ± 0.262** |
| | | no | 35.683 ± 0.164 | 75.776 ± 0.085 |
| | CPLLM-BioMedLM | yes | 35.494 ± 0.352 | **75.975 ± 0.214** |
| | | no | **35.714 ± 0.516** | 75.794 ± 0.194 |

Table 3: PR-AUC and ROC-AUC for CPLLM-Llama2 and CPLLM-BioMedLM, across three distinct medical tasks. Added Tokens column indicates whether additional tokens were incorporated into the pre-trained tokenizer, with "yes" signifying the inclusion of additional tokens and "no" indicating without additional tokens.

## 4 DISCUSSION

Our proposed CPLLM method outperformed the baselines on all three tasks across two different datasets. We used MIMIC-IV and eICU-CRD datasets to assess the model's ability to handle two different code types (ICD9 and ICD10) and two data types (homogeneous data from the same hospital in MIMIC-IV and multi-center data in eICU-CRD). CPLLM was superior to all baselines. CPLLM-Llama2 was the best model overall, and only for Chronic kidney disease did CPLLM-BioMedLM outperform CPLLM-Llama2, but only in terms of ROC-AUC. Using CPLLM-Llama2, we achieved PR-AUC relative improvements of 3.309%, 1.864%, and 7.588% over Med-BERT on the three tasks, and ROC-AUC relative improvements of 1.326% and 1.391% on the Adult respiratory failure and Acute and unspecified renal failure prediction tasks.

Unlike existing approaches that necessitate pre-training with medical concept sequences, our method eliminates the need for additional pre-training tasks. For instance, Med-BERT entails both MLM and LOS prediction tasks using patient sequences of medical concepts. Based on our findings and results, it's evident that LLMs possess the capability to adeptly represent sequential clinical data without the need for specific pre-training based on clinical sequences. Beyond that, our method can be used even without the LOS data of each patient's hospitalizations, which is required for Med-BERT pre-training. Sometimes, these data are not available, for example, when there is no hospitalization, but rather data collected among patients who visited a physician in outpatient settings, or when LOS data is not available like in claims data. Furthermore, during the fine-tuning training of CPLLM, it is not necessary to know which diagnoses were given in which visit but only the diagnoses as a sequence. This differs from Med-BERT, which relies on this information for fine-tuning. Notably, we achieved superior performance even without these specific details.

We found that including additional tokens in the LLM's tokenizer before fine-tuning improves the measurement of the prediction model in most cases. For instance, as Llama2 was not initially pre-trained on clinical data, supplementing it with missing description codes can enhance its understanding of the medical domain.

In the original Med-BERT paper, improvements over RETAIN were demonstrated in terms of ROC-AUC for three prediction tasks (Rasmy et al., 2021). We found as well that Med-BERT consistently outperformed RETAIN in all prediction tasks based on ROC-AUC. However, it's worth noting that, as previously mentioned, ROC-AUC may not be an optimal metric for imbalanced datasets (Davis & Goodrich, 2006). In contrast, when considering PR-AUC, Med-BERT exhibited superior performance compared to RETAIN in two out of three tasks, although it did not outperform RETAIN in

the prediction of Acute and unspecified renal failure (with PR-AUC values of 43.603% for RETAIN and 42.237% for Med-BERT), despite achieving a higher ROC-AUC than RETAIN.

Although our current implementation focuses on diagnosis input, our method offers the flexibility to incorporate medical concepts from any domain into the sequence with minimal adjustments to the prompt text. For example, medications, procedures, and more can be in the sequence together with the current diagnosis codes.

Another strength of our proposed method lies in its remarkable capacity to handle longer sequences compared to the current state-of-the-art models for structured EHR data. With maximum sequence lengths of 1024 tokens for CPLLM-BioMedLM and 4096 tokens for CPLLM-Llama2, our approach far surpasses the limitations imposed by Med-BERT and BEHRT (Li et al., 2020). Both Med-BERT and BEHRT are constrained by BERT's maximum of 512 tokens, which significantly restricts their ability to handle longer inputs (Devlin et al., 2018). Without the need for additional training, our method also handles longer sequences compared to Hi-BEHRT, which is specially trained and designed to handle sequences with a maximum of 1220 tokens (Li et al., 2022a).

**Limitations**: While our method demonstrates promising results in utilizing LLMs for clinical prediction tasks, it is important to acknowledge several limitations. While our method accommodates sequences of up to 4096 tokens for CPLLM-Llama2 and 1024 tokens for CPLLM-BioMedLM, our tests did not include exceptionally long sequences that could fully explore the implications of this extended token limit. That is because the datasets we used do not contain very long observations or many diagnoses of a single patient. Moreover, due to the greater number of parameters in LLMs, our method demands more computational resources, inference time, and training time. Specifically, CPLLM-Llama2 had a longer training time than Med-BERT. However, CPLLM-BioMedLM requires less training time compared to Med-BERT (3.1). That's because CPLLM-BioMedLM does not require additional pre-training, unlike necessity for MLM and LOS pre-training in Med-BERT.

In addition, in our method, there is a necessity to use a specific prompt, a requirement that does not apply to the baseline models. As a result, sometimes the prompt needs to be adapted according to a base model.

**Future work**: We hypothesize that combining a retrieval augmentation (Mialon et al., 2023; Hiesinger et al., 2023), with LLM can improve performance. This is because it allows to include general updated knowledge about the diseases that the patient has been diagnosed with in their medical history. Additionally, this approach can incorporate general knowledge and known risk factors into research on the disease we are trying to predict.

## 5    CONCLUSION

In this work, we presented CPLLM, a novel method for clinical disease prediction based on the clinical history of patients. We showed that CPLLM outperforms the state-of-the-art methods. CPLLM demonstrated superior performance across all three tasks on two datasets (MIMIC-IV and eICU-CRD). It handles ICD9 and ICD10 diagnosis codes and we showcased its robustness in dealing with homogeneous and multi-center data. Our method's advantage lies in eliminating the need for additional pre-training tasks, unlike Med-BERT. Furthermore, our method remains adaptable the length of stay data is unavailable, making it suitable for a broader range of healthcare scenarios, including those involving non-hospitalized patients. In addition, CPLLM's fine-tuning process requires patients' diagnoses as a sequence, without the need for which diagnoses were given in which visit. Notably, our method can handle much longer sequences than existing state-of-the-art models.

## 6    REPRODUCIBILITY

Our code is provided in the Supplementary Material . Implementation details can be found in the Experimental Setup section 3.1. To execute the baseline code, we used the source code published as part of the Med-BERT paper (Rasmy et al., 2021).

For our experiments, we used the MIMIC-IV v2.0 dataset (Johnson et al., 2020), accessible at https://physionet.org/content/mimiciv/2.0/, as well as the eICU-CRD multi-center dataset (Pollard et al., 2018), which can be found at https://physionet.org/content/eicu-crd/2.0/.

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
