# OpenReview forum: "CPLLM: Clinical Prediction with Large Language Models"
_ICLR.cc/2024/Conference — ICLR 2024 Conference Withdrawn Submission_

### Official Review · Reviewer_9sDQ · 2023-10-22

**Soundness:** 2 fair
**Presentation:** 2 fair
**Contribution:** 2 fair
**Rating:** 1
**Confidence:** 5

**Summary:**

This paper introduces a novel approach using a large language model (LLM) to predict next-visit diseases by leveraging temporal structured Electronic Health Record (EHR) data. This approach is termed CPLLM.

To develop CPLLM, the authors adopted pre-trained LLMs, such as LLAMA2, and fine-tuned them using the Quantized LORA technique. This fine-tuning process involved translating EHR data into text, allowing the model to interpret and process medical records in a more contextually rich manner.

For the purpose of validation, the authors benchmarked the performance of CPLLM against several prominent baselines, which include Logistic Regression, RETAIN, and Med-BERT. The results demonstrate that CPLLM outperforms these baseline models, underscoring its potential to enhance disease prediction based on EHR data.

**Strengths:**

The paper is generally well-structured and comprehensible. It presents experiments that demonstrate the efficacy of LLMs in disease diagnosis using textualized EHR data, coupled with vanilla instruction tuning.

**Weaknesses:**

- Method novelty is limited: The paper predominantly leans on existing techniques, specifically, LLM and LORA, in the context of a familiar task – disease diagnosis using Electronic Health Records (EHR). This raises questions about the originality of the work.

- Baselines are weak and not comprehensive enough: The selected baselines, which include Logistic Regression, RETAIN, and MedBERT, seem outdated relative to the current landscape of the domain. Not only do they represent older models, but their choice also misses out on potential recent advancements in disease diagnosis using EHR.

- Performance is not significant: The reported performance metrics suggest that the newer methods are only marginally better or even comparable to some of the baseline methods. For instance, both MedBERT and RETAIN demonstrated performances that were nearly on par with the basic Logistic Regression. This raises concerns about the effectiveness of the proposed approach and the tuning of the baseline models.

**Questions:**

- Table2: why all these methods performance look so similar? Why no standard deviation/confidence interval for LR?

- Novelty: what is the most prominent novelty the authors want to highlight in this paper?

---

### Official Review · Reviewer_i4m4 · 2023-10-26

**Soundness:** 1 poor
**Presentation:** 1 poor
**Contribution:** 1 poor
**Rating:** 1
**Confidence:** 4

**Summary:**

This paper introduces CPLLM for clinical prediction with large language models (LLMs). The authors propose to fine-tune LLMs with patients' diagnoses, modeled as the next diagnosis prediction task. The model is fine-tuned by ICU datasets, and evaluated on predicting three specific diseases from ICU datasets. There are marginal performance improvements over baselines like logistic regression. The authors have also shown that adding tokens such as "yes" improves the performance.

**Strengths:**

Modeling clinical diagnosis prediction as a pre-training task and fine-tuning both general and biomedical LLMs with it. Experimental results show performance improvement.

**Weaknesses:**

1. **Very marginal performance improvement**. Empirically, the model only outperforms logistic regression by about 1%. The performance improvement is very incremental, and besides that, I don't think the proposed methods can be favored more than LR in actual implementation in terms of inference latency and model interpretability.
2. **Lack of novelty and technical sophistication**. As cited and compared in the paper, there have been a lot of other methods that pre-train on the task of "next diagnosis prediction", with more technical sophistication and modalities beyond diagnosis codes. This paper only conducts superficial training on this task. I don't see there are any benefits of modeling this task as a language generation task compared to a time-series generation task.
3. **No in-depth analysis**. There are no case studies or error analyses provided, so we don't know why the model is better and where the mistakes are made.
4. **Narrow applications**. The model is pre-trained by ICU datasets and is evaluated only on predicting three specific diseases from ICU cohorts. The selection of the diseases seems arbitrary to me, and the generalizability of the proposed method to other diseases is not clear.

**Questions:**

1. What's the difference between "Next diagnosis prediction" and "Next visit diagnosis prediction"?

---

### Official Review · Reviewer_qHnV · 2023-10-28

**Soundness:** 2 fair
**Presentation:** 3 good
**Contribution:** 2 fair
**Rating:** 3
**Confidence:** 5

**Summary:**

This work presents an approach for developing clinical prediction models that operate over structured electronic health record (EHR) data using general-purpose large language models (LLMs) without any pretraining with EHR data. The approach represents patient medical history as sequences of historical diagnoses and their natural language descriptions to predict the presence of diagnosis codes at the next observed visit. The LLMs are separately fine-tuned for each prediction task using a task-specific prompt and parameter-efficient fine-tuning methods using labeled data from the target EHR database. An empirical study is conducted with the MIMIC-IV and eICU-CRD databases for three disease prediction tasks and the method compared to logistic regression, RETAIN, and Med-BERT.

**Strengths:**

The central claim and value proposition for this work is strong. Being able to develop performant clinical prediction models by fine-tuning general-purpose large language models could greatly reduce the need for large-scale foundation models pretrained on structured electronic health record data.

**Weaknesses:**

* The empirical evaluation would be stronger if the approach was evaluated in settings where only a no or few examples from the target EHR database are available. This could involve settings with zero-shot and few-shot prompting and with or without any finetuning of the base model.
* The use of only diagnosis codes significantly limits the contribution of this work relative to the prior literature on machine learning with EHRs. It is standard in the field to consider diagnoses, procedures, medication, laboratory tests and results, vital signs, and clinical notes, among other observations (see e.g., Rajkomar 2018, Wang 2020). While the use of diagnosis codes alone serves as an interesting proof-of-concept, it is a limited enough setting that I would be hesitant to generalize the conclusions of this work to the more generic setting.
* The procedure for identifying labeled examples introduces a subtle selection bias that limits alignment of the selected patient cohort with a real-world distribution. The issue lies in that patients positive for a disease are selected based on the presence of disease at any time, with the time of prediction selected as the previous time point, and patients negative for the disease selected at a randomly selected time point. In a real-world deployment, the time of prediction cannot depend on knowledge of the future development of disease. A random sample from the real-world target population would be sampled IID from the set of time points for which the diagnosis of interest had not already occurred in a patient’s medical history. The models fit in this work will not generalize to this ideal target population due to the way that positive examples are selected by looking into the future and selecting the prior timestep to make a prediction. It is also not a matter of a simple adjustment for a base rate shift because the structure of the selection affects the conditional distribution of the covariates given the label (e.g. the selection bias violates the label shift invariance).

References
* Rajkomar, Alvin, et al. "Scalable and accurate deep learning with electronic health records." NPJ digital medicine 1.1 (2018): 18.
* Wang, Shirly, et al. "Mimic-extract: A data extraction, preprocessing, and representation pipeline for mimic-iii." Proceedings of the ACM conference on health, inference, and learning. 2020.

**Questions:**

* In prior work (e.g., Steingberg 2020), the value of foundation model pretraining over from-scratch baselines is more stark in the few-shot setting, with the magnitude of the differences narrowing when a large amount of data from the target task is available for fine-tuning. The current work effectively only evaluates in a setting where a large amount of target data is available and the performance margins are small, consistent with the prior work. Additional experiments involving zero/few-shot prompting with or without fine-tuning would help build understanding as the potential pros/cons of using an LLM vs. a model pretrained with EHR data in different data-availability regimes.
* Do the baselines also only process diagnosis codes or do they have access to other data types? I am particularly interested in the RETAIN and Med-BERT results given that the direct use of those methods in other work processes additional data types (e.g. procedures, medications, etc). Does this affect the validity and generalizability of conclusions about that baselines and CPLLM?
* The experiments would be stronger if re-designed to correct for the selection bias issue mentioned in the weaknesses section. To be clear, the timepoints used for prediction should not be selected based on knowledge of the future.
* Given the emphasis given to arguing that the approach generalizes to different contexts (e.g. not inpatient/hospital data), the work would be strengthened with some evaluation on outpatient or claims data.

References
* Steinberg, Ethan, et al. "Language models are an effective representation learning technique for electronic health record data." Journal of biomedical informatics 113 (2021): 103637.

---

### Official Review · Reviewer_M1Ux · 2023-10-31

**Soundness:** 3 good
**Presentation:** 2 fair
**Contribution:** 1 poor
**Rating:** 3
**Confidence:** 4

**Summary:**

The authors propose Clinical Prediction with Large Language Models, which performs well on next disease on structured EHR data.

**Strengths:**

- The paper is quite clear and relatively free of grammatical errors. Tables and figures are also clear.

**Weaknesses:**

- The novelty of applying LLMs for this task is not clear, as instruction tuning of an LLM for a classification task is now a common approach, despite the good performance.
- There is a major concern regarding the chosen baselines. For example, there are many other baselines implemented in PyHealth (https://pyhealth.readthedocs.io/) on similar clinical predictive tasks.
- The tasks evaluated are not comprehensive, as disease prediction is limited to only 3 tasks. - The proposed method does not seem to offer a large improvement over logistic regression

**Questions:**

- I believe LLMs, due to their power, can support a much larger variety of tasks, such as Readmission Prediction, Mortality Prediction, Length of Stay Prediction, and more as seen on PyHealth. Would it be possible to run these as well?